# Quantitative measurement of antibiotic resistance in *Mycobacterium tuberculosis* reveals genetic determinants of resistance and susceptibility in a target gene approach

**The CRyPTIC Consortium***

The World Health Organization has a goal of universal drug susceptibility testing for patients with tuberculosis. However, molecular diagnostics to date have focused largely on first-line drugs and predicting susceptibilities in a binary manner (classifying strains as either susceptible or resistant). Here, we used a multivariable linear mixed model alongside whole genome sequencing and a quantitative microtiter plate assay to relate genomic mutations to minimum inhibitory concentration (MIC) in 15,211 *Mycobacterium tuberculosis* clinical isolates from 23 countries across five continents. We identified 492 unique MIC-elevating variants across 13 drugs, as well as 91 mutations likely linked to hypersensitivity. Our results advance genetics-based diagnostics for tuberculosis and serve as a curated training/testing dataset for development of drug resistance prediction algorithms.

*Mycobacterium tuberculosis* (Mtb) caused an estimated 10 million new cases of tuberculosis (TB) and 1.4 million deaths in 2019[1]. Of particular concern are the estimated 465,000 rifampicin resistant (RR) cases, 78% of which were multi-drug resistant (MDR, resistant to both rifampicin and isoniazid)[1]. Drug resistance poses two major challenges to the successful treatment of TB, as it is both underdiagnosed (only 38% of RR/MDR cases in 2019)—leading to under-treatment—and has poor treatment success rates even when identified (57% globally in 2019)[1]. Despite attempts to move to shorter and all-oral MDR TB regimens using new drugs, most patients are still receiving toxic regimens that decrease patient adherence[1,2]. Collectively, the failure to identify and successfully treat these cases leads to onward transmission and amplification of drug resistant strains

The WHO has identified better diagnosis and treatment of drug resistant tuberculosis as a key part of the global tuberculosis eradication strategy[1]. Rapid genetics-based diagnostic tools, such as GeneXpert, have been widely adopted as they are faster and cheaper than traditional culture-based diagnostic susceptibility testing (DST). However, outbreaks caused by drug-resistant strains with mutations not detected by such assays reveal the importance of developing assays that include a wider range of resistance determinants[3]. Some

approaches incorporate whole-genome sequencing (WGS) or targeted next generation sequencing to identify all possible resistant variants and recently these methods have proven to be capable of replacing culture-based DST for the first line drugs; however, implementation of this technology is not yet feasible globally due to cost and technical expertise constraints[4–6].

Most current culture and genetics-based DST approaches generate binary results—"resistant" or "susceptible"—and thus fail to consistently report elevations in minimum inhibitory concentration (MIC) below or around the critical concentration[7]. These sub-threshold elevations in MIC may nevertheless be clinically meaningful, as the combination of significant interpatient pharmacokinetic variability and elevated MICs predisposes Mtb strains to development of higher-level resistance, risking treatment failure and worse patient outcomes[8,9]. A binary system also hampers the wider implementation of informed high-dose regimens which have been trialed to extend the clinical utility of relatively less toxic and more widely available drugs such as rifampicin and isoniazid[10–12]. While some previous efforts have attempted to use quantitative MICs to identify these lower-level resistance variants, they were limited by smaller sample sizes and combined heterogenous methods of resistance determination[13].

---

Additionally, relatively few studies have had adequate sample sizes to investigate drugs such as bedaquiline, linezolid, clofazimine and delamanid that are poised to become the new "front-line" drugs for the MDR-TB treatment.

To resolve these issues, we performed WGS and determined the MICs of 13 drugs for 15,211 Mtb isolates selected from patient samples gathered from 23 countries over five continents using a previously validated microtiter plate[14]. This data covers all first-line drugs (except pyrazinamide), as well as eight drugs from the new MDR-TB treatment guidelines (all Group A, one Group B, and four Group C)[15]. Overall, we identify 492 unique mutations that are associated with elevated MICs across 13 drugs as well as mutations that are associated with increased susceptibility to bedaquiline, clofazimine, and the aminoglycosides. The results serve as guides for pharmacokinetic and dosing studies to extend the clinical utility of less toxic and more widely available drugs for the treatment of drug-resistant tuberculosis, as well as help to improve the design of genetics-based rapid diagnostics for MDR-TB and the recently published WHO genetic catalog for tuberculosis[16]. They also provide a large, quality-controlled dataset for development of drug resistance prediction algorithms using machine-learning and other approaches.

## Results

### Dataset description
Bacterial isolates were collected from patient samples from 23 different countries and were over-sampled for drug resistance. Of the 15,211 isolates included in the initial CRyPTIC dataset, 5541 were phenotypically susceptible to isoniazid, rifampicin, and ethambutol, 5602 were isoniazid resistant, 5261 were rifampicin resistant, and 4,125 were multidrug-resistant (MDR, resistant to both rifampicin and isoniazid) based on previously published epidemiological cutoffs (ECOFF, MIC that encompasses 99% of wild type) for the microtiter plates used in

this study[17]. Binary phenotypic resistance to the newer drugs was observed at a lower prevalence, with 71 bedaquiline resistant isolates, 106 clofazimine-resistant isolates, 76 linezolid resistant isolates, and 85 delamanid resistant isolates (Supplementary Data 1). Isolate lineages were determined using a published SNP-based protocol from WGS data and the lineage distribution across countries reflects previously described phylogeographic distributions[18–20]. Five out of eight major lineages of Mtb were represented in the dataset, with most isolates mapping to L4 (6572 isolates) and L2 (5598 isolates), while L3 (1850), L1 (1150), and L6 (6) comprised the remainder. A complete description of the CRyPTIC dataset and determination of the ECOFFs have been previously published (also see Methods)[17,21]. After the removal of isolates due to errors in phenotyping and sequencing across sites, the final genotype/phenotype intersection for all drugs was ~12,350 isolates (Fig. 1).

### Genetic resistance determinants in Mycobacterium tuberculosis
Previous studies have shown that the majority of genetic determinants of resistance to most anti-tuberculosis drugs are related to a relatively small number of genes[6,22]. We thus employed a candidate gene approach and restricted our investigation of genomic variation to previously identified genes and the 100 bp directly upstream of each gene for each drug (Table 1). All unique variation in the target genes and upstream regions (SNPs, both synonymous and nonsynonymous, as well as insertions and deletions <50 base pairs in length) that occurred in isolates with matched high-quality phenotypic data was included in a separate multivariable linear mixed model controlling for population structure and technical variation between sites for each drug, after removing isolates with evidence for mixed allele calls at sites previously identified as resistant (e.g., *rpoB* S450X, Methods). Final sample sizes per drug ranged from 6681 for moxifloxacin to 10,042 for rifabutin (mean sample size 8353, Fig. 1, Methods). Most

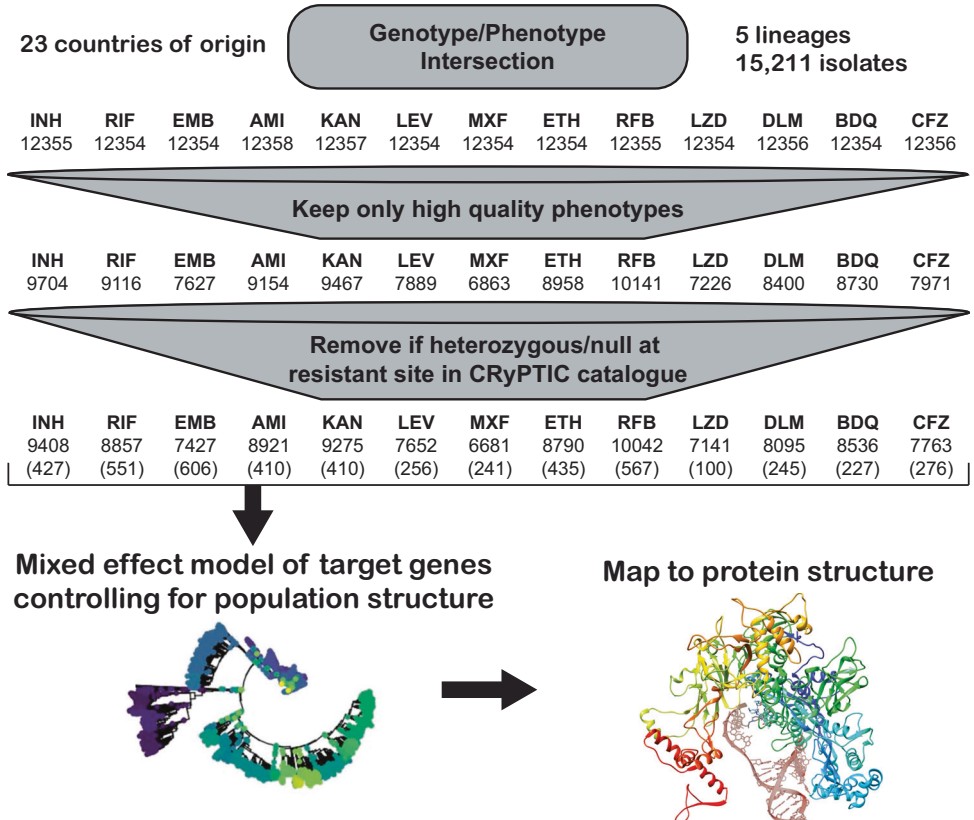

**Fig. 1 | Data flow and sample sizes for CRyPTIC MIC models.** Numbers in brackets represent the number of variables (mutations plus lineage and site effects) included in the final model.

**Table 1 | Candidate genes used in this study**

| Drug | Abbrev. | Candidate genes |
|---|---|---|
| Isoniazid | INH | *katG, fabG1, inhA, ahpC, ndh, kasA, Rv1258c, Rv2752c* |
| Ethionamide | ETH | *ethA, ethR, fabG1, inhA, mshA, Rv3083, Rv0565c* |
| Rifampicin | RIF | *rpoA, rpoB, rpoC, rpoZ, Rv2752c* |
| Rifabutin | RFB | *rpoA, rpoB, rpoC, rpoZ, Rv2752c* |
| Ethambutol | EMB | *embA, embB, embC, embR, rmlD, iniA, iniC, manB, ubiA* |
| Amikacin | AMI | *rrs, eis, ccsA, whiB6, whiB7, aftB, fprA* |
| Kanamycin | KAN | *rrs, eis, ccsA, whiB6, whiB7, aftB, fprA* |
| Levofloxacin | LEV | *gyrA, gyrB* |
| Moxifloxacin | MXF | *gyrA, gyrB* |
| Bedaquiline | BDQ | *atpE, Rv0678, mmpL5, mmpS5, pepQ, Rv3249c* |
| Clofazimine | CFZ | *Rv1979c, pepQ, Rv0678, mmpL5, mmpS5, Rv3249c* |
| Linezolid | LZD | *rplC, rrl, Rv3249c* |
| Delamanid | DLM | *ddn, fgd1, fbiA, fbiB, fbiC, fbiD, Rv3249c* |

isolates had less than five nonsynonymous mutations across all target genes for each drug (Supplementary Data 2).

Across thirteen drugs, 584 mutations in 40 genes (out of 4,778 mutations and 50 genes tested) were found to have independent effects on MIC after correction for multiple testing (Benjamini-Hochberg correction, false discovery rate <0.05, Fig. 2, Table 1, Supplementary Data 3). Ethionamide had the most unique variants associated with reduced susceptibility (163), while linezolid had the least (8). Effect sizes were measured in log2MIC (where an increase in 1 log2MIC was equivalent to a doubling of the MIC) and positive effects for estimates derived from at least three observations ranged from a 0.22 increase in kanamycin log2(MIC) for *rrs* c492t to a 10.1 increase in isoniazid log2(MIC) for *katG* W477Stop. To facilitate comparison with previously published ECOFF values, we report mutational effects relative to the difference between the ECOFF MIC and the baseline MIC calculated by the model for each drug. Thus, if a mutation is associated with an effect larger than the ECOFF minus baseline, it is associated with an increase in resistance that would be above what is considered wildtype on the plate. Multiple promoter mutations were implicated in resistance to isoniazid, ethionamide, amikacin, kanamycin, and ethambutol (Fig. 2B). The effects of promoter mutations varied widely, with mutations upstream of *eis* and *embA* being almost exclusively associated with sub-ECOFF elevations in MIC for amikacin and ethambutol respectively, while most promoter mutations for the isoniazid and ethionamide-related *fabG1* resulted in MICs above the ECOFF[17]. While a prior study found that common promoter mutations tended to be associated with lower-level resistance than their corresponding common gene-body counterparts (e.g., *fabG1* c-15 vs *inhA* I21), we found that mutations affecting coding sequences vs mutations affecting promoters/intergenic regions were only associated with significantly different effects on MIC for isoniazid, ethambutol, and kanamycin (Supplementary Data 4)[13]. In fact, we found that the widespread *fabG1* c-15t promoter mutation was associated with higher-level and equivalent-level resistance to its gene body counterparts *inhA* I21V and I21T respectively (Fig. 2B, Wald test for equality of coefficients $p = 0.0006$, $p = 0.24$, respectively). Resistance-associated promoter mutations were enriched in the region around each gene's respective −10 element, which is consistent with the essentiality of the −10 hexamer and increased variability around the −35 position in mycobacterial promoters (Fig. S1, ±5 nucleotides, Mantel-Haenszel common OR = 4.5, $p = 0.0007$)[23,24]. Multiple insertion/deletion mutations were associated with resistance to isoniazid, rifampicin, rifabutin, ethionamide, ethambutol, bedaquiline, clofazimine, and delamanid (Supplementary Data 3, Fig. 2). Homoplastic mutations (multiple evolutionarily independent occurrences) were more likely to be

associated with resistance for all drugs except amikacin, kanamycin, clofazimine, linezolid, and delamanid (Woolf test for homogeneity of ORs $p = 0.0004$, Supplementary Data 5). The relative lack of homoplasy in the newer drugs may reflect the lower prevalence of resistant isolates observed for these drugs as opposed to lack of convergent evolution.

One notable advantage of quantitative MIC measurements is that they also enable investigation of variants associated with MIC decreases. We identified 63 increased susceptibility-associated mutations (with at least three occurrences) whose effect sizes ranged from −4.3 rifampicin log2(MIC) for *Rv2752c* H371Y to −0.23 kanamycin log2(MIC) for *eis* V163I (Fig. 2A, Supplementary Data 6). Eight of these mutations were homoplastic with at least three independent occurrences, which raises the intriguing possibility of a selective pressure for mutations associated with increases in drug susceptibility; however, this remains to be verified experimentally.

**First-line drugs**

Rifampicin is a critical first line drug and resistance to it is almost entirely mediated by mutations within an 81-base pair region of the *rpoB* gene (rifampicin resistance determining region, RRDR). Most molecular assays target mutations in this region for rapid prediction of rifampicin resistance, however, mutations outside this region have been associated with outbreaks[25,26]. We identified 35 mutations in *rpoB* occurring at least three times whose effects collectively ranged from 1.0 to 9.0 increases in log2MIC (Fig. 3A). Notably, seven unique resistance-associated mutations occurred outside the RRDR, at positions V170, Q172, I491, and L731; however, only V170F was associated with high-level resistance (8.37 increased log2MIC). Although disparate in primary sequence from the RDRR, positions V170, Q172 and I491 are all near the drug-binding pocket structurally (Fig. 3B). Interestingly, a homoplastic in-frame deletion 12 bp in size in the RRDR was also associated with rifampicin resistance (Fig. 3C, Supplementary Data 3). Several types of insertion/deletion mutations in the RDRR have previously been reported, although they are rare, consistent with their greater structural consequences for the essential RNA polymerase[27].

Prior studies have identified seven "borderline" mutations in *rpoB* (L430P, D435Y, H445L, H445N, H445S, L452P, and I491F) for rifampicin; resistant isolates with these mutations are often missed by phenotypic methods such as the Mycobacterial Growth Indicator Tube (MGIT), possibly due to slower growth rates, which has led to a reduction in the critical concentration for MGIT in the latest WHO guidelines[28–30]. These mutations' MICs range on the plate from 5.1 log2MIC for H445L to 2.3 log2MIC for L430P (rifampicin ECOFF minus baseline MIC = 3.3, Supplementary Data 3). Here, we identify thirteen additional *rpoB* mutations independently associated with elevated MICs that are less than 5.1 log2MIC (8/13 located in the RDRR, Supplementary Data 3). Sixteen *rpoB* mutations in total were independently associated with elevated MICs at or below the rifampicin ECOFF, including *rpoB* L430P, a variant that has been successfully treated with a high dose rifampicin-containing regimen clinically[31]. Several *rpoB* positions (Q432, D435, H445) harbored both high and low-level resistance-associated alleles, while others (L430, L452, I491) were associated exclusively with lower-level resistance regardless of the amino acid substitution (Fig. 3B, C orange and yellow shading respectively). Mapping these mutations onto the rpoB protein structure revealed that high-level resistance often involves disruption of the interactions with the rigid napthol ring while mutations at positions that contact the ansa bridge had more variable effects, potentially due to increased structural flexibility in this region of the drug (Fig. 3B). Low-level resistance mutations often co-occurred with other low-level resistance mutations, producing high-level resistance additively (Fig. S4).

Rifabutin (a structural analog to rifampicin) is associated with a lower ECOFF (2.2 vs 3.3 log2MIC after subtraction of baseline) and mutations in *rpoB* were associated with lower elevations in rifabutin

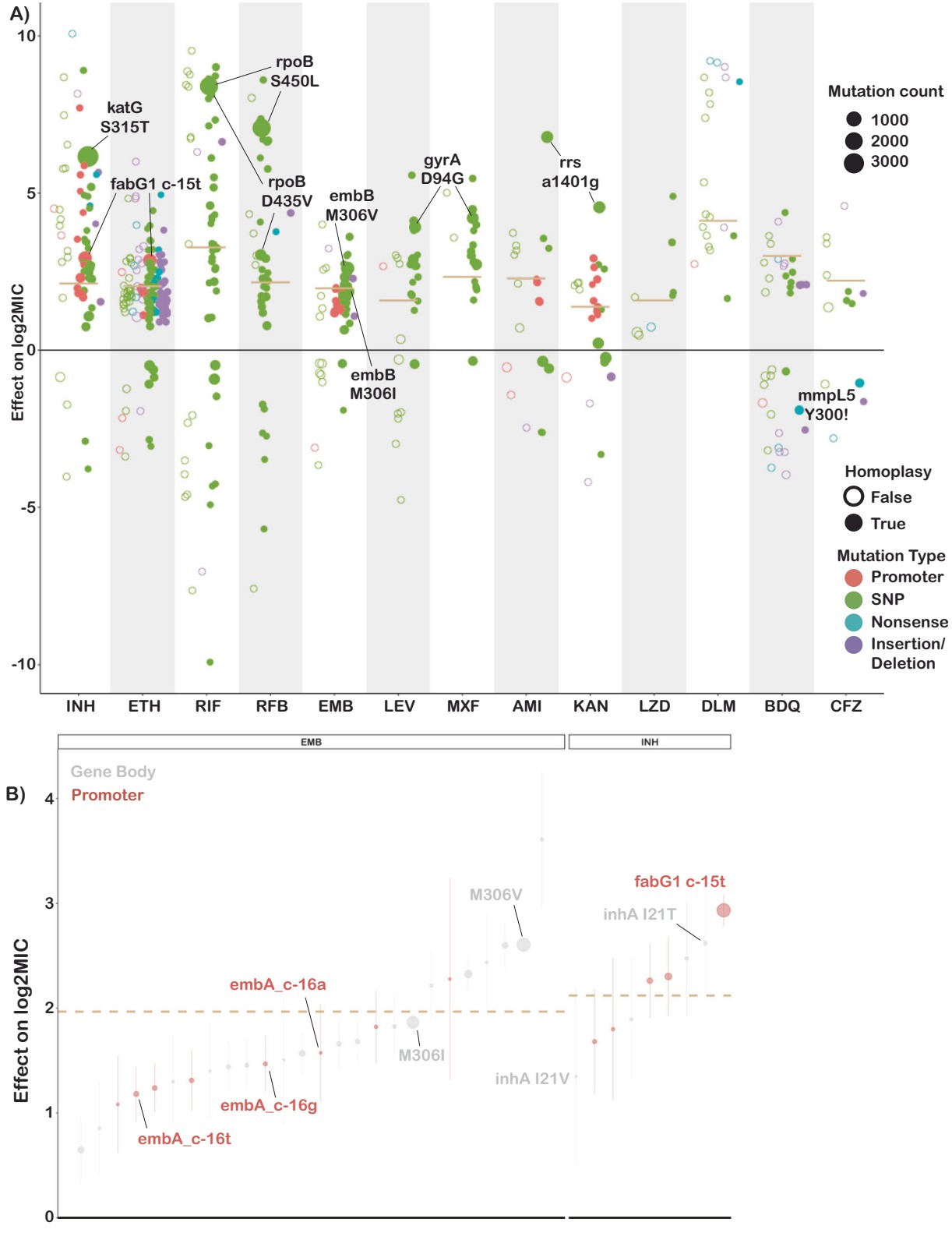

**Fig. 2 | Variation in effect size by mutation type and drug. A** Effects on log2MIC for the 540 variants significant after false discovery rate correction using the Benjamini-Hochberg method. Mutation types are delineated by color. Homoplastic mutations are shown as solid circles. ECOFF (minus baseline MIC) is shown as tan line. Common resistance mutations are highlighted. **B** Comparison of effects on log2MIC for promoter and corresponding gene body variants for ethambutol (EMB) and isoniazid (INH). ECOFF (minus baseline MIC) is shown as a tan line. Points are the mean effect in the interval regression model with error bar representing the 95% confidence interval. Exact effects, sample sizes and *p* values are provided in Supplementary Data 3.

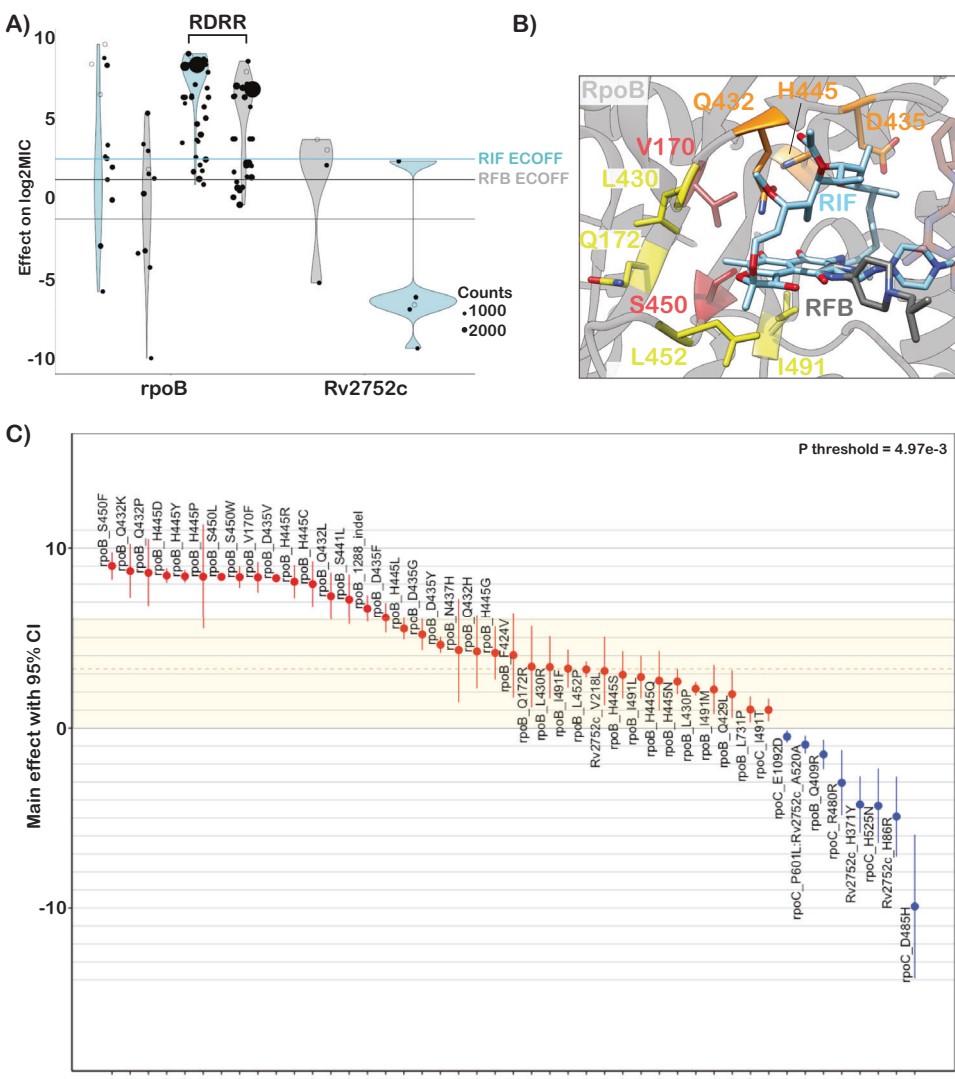

**Fig. 3 | Heterogenous effects of *rpoB* mutations on rifampicin resistance.**
**A** Mean effects of target gene variants on rifampicin (blue) and rifabutin (gray) log2MIC. ECOFFs (minus baseline MICs) are highlighted as lines. **B** Rifampicin (blue) and rifabutin (gray) bound to rpoB with resistance-associated variants highlighted (red-high, orange-variable, yellow-low). **C** Mean effects on rifampicin

MIC of mutations in *rpoB* with error bars representing 95% confidence interval. Exact sample size for each mutation is shown at the bottom of panel B. Colored shading highlights "borderline" variants. P-threshold is the value reaching significance after Bejamini-Hochberg correction for multiple testing. Sample sizes and p-values for each mutation effect are provided in Supplementary Data 3.

MIC compared to rifampicin MIC (paired Wilcoxon $p = 3.7e\text{-}9$, Fig. 3A, Supplementary Data 3). Interestingly however, all structural features contacted by these mutations were shared between rifampicin and rifabutin (Fig. 3B). A single mutation, *rpoB* Q409R ($n = 24$, $p = 5.0e\text{-}3$ after Benjamini-Hochberg (BH) correction), was associated with decreased rifampicin and rifabutin MICs; interestingly, this mutation has been proposed as a compensatory mutation that may alter the rate of transcription initiation and resulting transcription efficiency for isolates that harbor other RDRR mutations[32].

Resistance to isoniazid is mediated primarily through loss-of-function mutations in the prodrug-converting enzyme *katG*, with canonical high-level resistance caused by the S315T mutation, which was associated with a 6.2 log2 increase in MIC (Fig. 4A, compared to 2.1 log2MIC ECOFF minus baseline). Not all *katG* mutations were associated with high-level resistance, nearly half (15/31) being associated with increases in MIC at or below the ECOFF. No mutations likely to result in severe loss of function were associated with sub-ECOFF resistance, supporting the consensus of treating presumptive loss-of-function mutants in *katG* as resistant. The other canonical isoniazid-

related genes, *inhA* and *fabG1*, tended to be associated with lower-level resistance, with 4/6 and 5/6 mutations associated with sub-ECOFF increases in MIC, respectively (Fig. 4A, Supplementary Data 3). While *fabG1* L203L was previously the only synonymous mutation known to be associated with resistance to isoniazid, here we identify a synonymous mutation in the first codon of *katG* that confers high-level resistance to isoniazid, potentially by reducing the rate of translation initiation and subsequent production of katG enzyme required for activation of isoniazid, although this is a mechanistic hypothesis that requires biochemical confirmation (4.5 log2MIC, $n = 3$, $p = 1.4e\text{-}8$ after Benjamini-Hochberg (BH) correction, Supplementary Data 3).

Most isoniazid resistance-associated mutations in *katG* occurred in the N-terminal lobe responsible for heme-binding and pro-drug conversion (Fig. 4B). Most isolates harbored variation at position S315, located in the primary isoniazid-binding pocket on the δ edge of the heme; interestingly however, another cluster of resistance-associated mutations occurred in the helix made up of residues 138–155. Some structural evidence exists for promiscuous isoniazid binding at this site and mutations of this region in *Escherichia coli* cause reduced

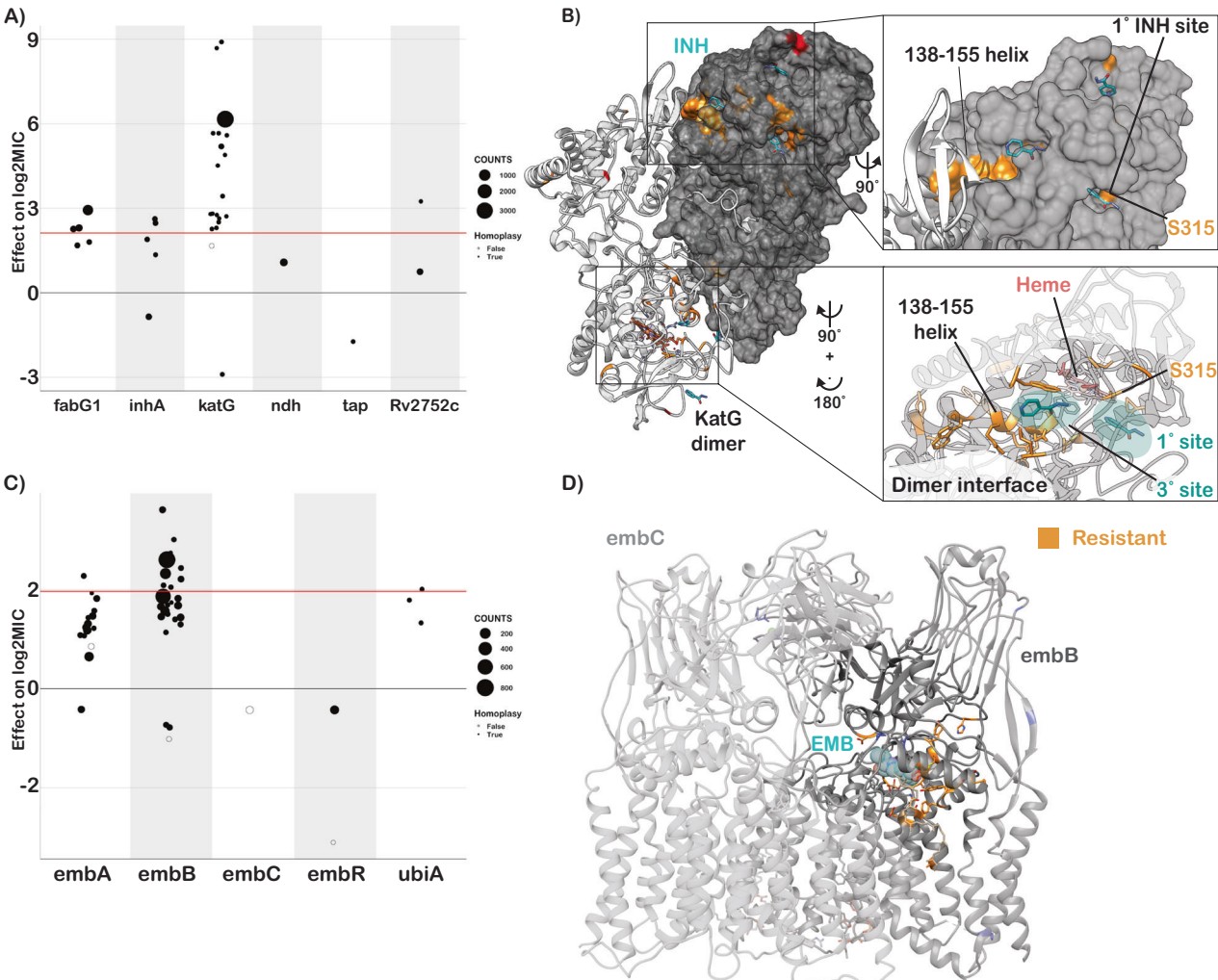

**Fig. 4 | Resistance to isoniazid and ethambutol is a multi-gene phenomenon.**
**A** Independent effects of variants in target genes on isoniazid log2MIC. ECOFF (minus baseline MIC) is denoted in red. **B** KatG dimer with isoniazid (blue) modeled and resistance-associated positions highlighted in orange. **C** Independent effects of variants in target genes on ethambutol log2MIC. ECOFF (minus baseline MIC) is denoted in red. **D** EmbA-embB complex bound to ethambutol (blue) with resistant mutations highlighted in orange. Sample sizes and *p* values for all effects are provided in Supplementary Data 3.

catalase/peroxidase activity and heme binding; however the precise mechanism of effect of these mutations in Mtb is unknown[33,34]. Intriguingly, one mutation in this region, *katG* S140N, was associated with decreased isoniazid MIC ($n = 9$, $p = 5.4e\text{-}4$ after BH correction, Fig. 4B).

Non-canonical isoniazid resistance-associated variants were identified in *ahpC*, *ndh*, and *Rv1258c* (*tap*) (Fig. 4A). Mutations in *ahpC* were associated with increased MICs; however, these mutations almost always co-occurred with mutations in canonical isoniazid genes and investigation of the interaction between these co-occurring mutation pairs revealed that *ahpC* mutations did not result in additive resistance, consistent with their proposed compensatory role (Fig. 4A). Further investigation of these apparent discrepant isolates using an improved version of the Clockwork variant calling pipeline that detected deletions larger than 50 bp identified nine isolates with apparent resistance-associated *ahpC* mutations that harbored large deletions in *katG* not reported in the original variant set used for the model. Thus, the apparent effect of these mutations is likely due to isolates with undetected mutations in the canonical resistance genes as opposed to a bona fide individual effect on isoniazid MIC by mutations in *ahpC*. Several recent genome-wide association studies (GWAS) have implicated mutations in the ribonuclease/beta-lactamase *Rv2752c* in resistance and tolerance to both rifampicin and isoniazid; however, they also identified convergent mutations in drug-

susceptible strains[13,35]. While we identified nine nonsynonymous mutations with significant effects on log2MIC, only one, V218L, was shared between isoniazid and rifampicin, causing a 3.2 elevation in log2MIC for both drugs (Supplementary Data 3). Only one other *Rv2752c* variant was associated with elevated rifampicin MICs, while four variants in this gene were associated with elevated isoniazid MICs (Fig. 4A).

Canonical ethambutol resistance is mediated by mutations in *embA* or *embB*. We identified 45 variants, 12 in the *embC-embA* intergenic region, five in *embA*, and 28 in *embB*, that were independently associated with elevated ethambutol MICs (Fig. 4C). Mutations in the *embC-embA* intergenic region have been proposed to upregulate production of embA and embB by altered promoter structure. Most *embC-embA* intergenic variants were in the upstream region from −16 to −8, however three were located upstream around the −35 element (Fig. S1). All *embC-embA* intergenic and *embA* gene body mutations were associated with MIC increases below the ECOFF (EMB ECOFF = 2 log2MIC minus baseline, Fig. 4C, Supplementary Data 3). Interestingly, 22/28 mutations in *embB* were also associated with sub-ECOFF increases in MIC, including the canonical *embB* M306I. Low-level resistance mutations often co-occurred, resulting in high-level additive resistance, consistent with previous studies (Supplementary Data 6)[36]. Mutations associated with resistance in *embB* were clustered around

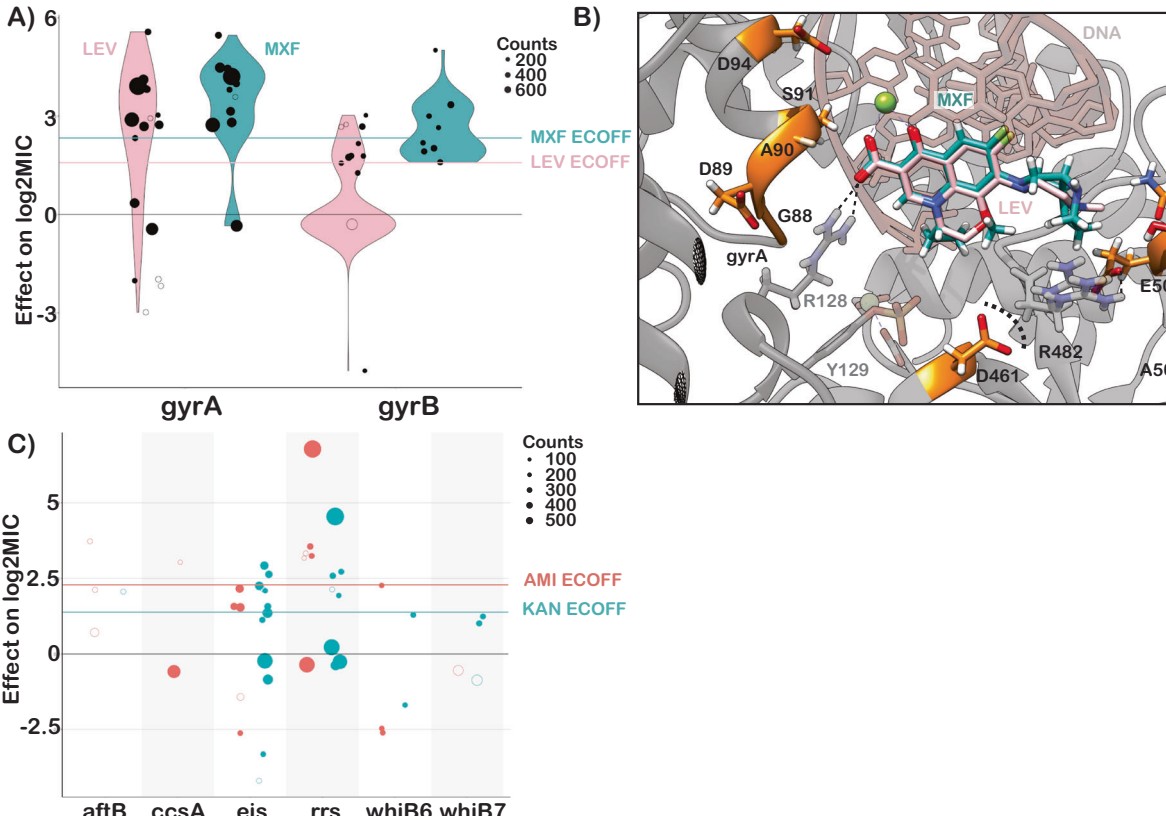

**Fig. 5 | Resistance to second line drugs. A** Effects of mutations in *gyrA* and *gyrB* on levofloxacin (pink) and moxifloxacin (green) log2MIC. **B** Structural mapping of fluoroquinolone resistance-associated variants reveal that majority lie within 10 Å of the drug binding site. Positions gyrB R446 and S447 are not shown. **C** Effects of mutations in aminoglycoside target genes on amikacin and kanamycin log2MIC. ECOFFs (minus baseline MIC) are shown for comparison. Sample sizes and *p* values for all effects are provided in Supplementary Data 3.

the drug-binding pocket (Fig. 4D)[37]. We also identified resistance-associated variants in *embC* and *ubiA*, although these occur less frequently and require further validation.

## Group A and B MDR drugs

The principal mechanism of resistance to fluoroquinolones is mutations in either subunit of DNA gyrase (*gyrA* or *gyrB*). We identified 22 mutations (12 *gyrA*, 10 *gyrB*) and 19 mutations (10 *gyrA*, 9 *gyrB*) that were independently associated with increased levofloxacin and moxifloxacin MICs respectively (Fig. 5A). Resistance-associated mutations in *gyrB* occurred without an accompanying *gyrA* mutation ~65% of the time (29/44 isolates LEV, 35/54 isolates MXF) but were associated with lower overall−and in some cases sub-ECOFF−changes in MIC (LEV ECOFF = 1.6 log2MIC, MXF ECOFF = 2.3 log2MIC, minus baseline, Fig. 5A, Supplementary Data 3, 6). Most mutations associated with increased fluoroquinolone MICs were within 10 Å of the drug binding pocket (Fig. 5B). Intriguingly, two positions−gyrB R446 and gyrB S447−each harbored two unique resistance-associated missense mutations despite being over 25 Å from the bound fluoroquinolone. Both residues make contacts with the gyrB protein backbone at positions 473–475, suggesting they may exert an allosteric effect by either influencing protein folding and/or the position of residues (notably D461 and R482) that make up part of the fluoroquinolone binding pocket (Fig. 5B). Interestingly, while gyrB E501D was associated with resistance 1 log2MIC above the moxifloxacin ECOFF, it did not cause a similar elevation for levofloxacin (only 0.1 log2MIC above ECOFF), consistent with previous studies[7,38,39]. We speculate this could be due to alteration in the coordination of gyrB R482−which must shift to accommodate the bulkier side group of moxifloxacin−although this remains to be shown experimentally (Fig. 5B).

While initial studies on bedaquiline and clofazimine resistance highlighted *atpE* (bedaquiline), *pepQ*, *Rv0678*, and *Rv1979c* as mediating resistance, surveillance of clinical samples has revealed the importance of the efflux mechanism mediated by the *mmpL5* membrane transporter, which is controlled by the transcriptional regulator *Rv0678*. Consistent with this, we identified sixteen and four mutations in *Rv0678* that were associated with elevated bedaquiline and clofazimine MICs, respectively, of which four were shared (Fig. S2, Supplementary Data 3). We also identified two *mmpL5* mutations that were associated with increased MICs for each drug which were not shared between the two drugs. Finally, we identified both the *atpE* E61D (*n* = 3) drug binding site mutation associated with bedaquiline resistance and two mutations in *Rv1979c* associated with clofazimine resistance. No mutations in *pepQ* were associated with resistance to either drug. Importantly, five unique nonsense and frameshift mutations in *mmpL5* increased susceptibility to bedaquiline by −1.9 to −4.0 log2MIC, of which one, *mmpL5* Y300Stop, was also shared with clofazimine (Fig. 2A). Inactivating mutations in *mmpL5* abrogated resistance mediated by co-occurring *Rv0678* mutations, consistent with a hypothesis proposed by a prior study[40].

Resistance to linezolid is mediated by mutations in *rplC* and *rrl*, which tend to cause higher- and lower-level resistance, respectively. We identified the classical *rplC* C154R (*n* = 43) mutation and five variants in *rrl* associated with elevated linezolid MICs (Fig. S2, Supplementary Data 3).

## Group C MDR drugs

Aminoglycoside resistance is canonically mediated by mutations in the 16s rRNA encoded by *rrs*. We identified five and six mutations in *rrs* that were independently associated with elevated MICs for amikacin and

kanamycin respectively (Fig. 5C). Multiple promoter mutations in *eis* were associated with elevated MICs to kanamycin (7) and amikacin (3). Interestingly, *eis* promoter mutations were associated with sub-ECOFF elevations in MIC for amikacin, while being associated with elevations in MIC comparable to *rrs* mutations for kanamycin (AMI ECOFF = 2.3 log2MIC minus baseline). A deletion in *eis* leading to loss of function was also associated with increased susceptibility to kanamycin, consistent with an epistatic interaction abrogating the resistance gained from *eis* overproduction. Variants in *aftB*, *ccsA*, *whiB6* and *whiB7* were also associated with elevated MICs for at least one aminoglycoside, however they were infrequent and require further investigation (Fig. 5C and Supplementary Data 3).

Ethionamide is a prodrug that is activated by the mono-oxygenases *ethA*, *mymA* (*Rv3083*), and *RvO565c*[41]. More variants (135) were associated with increased ethionamide resistance than any other drug, with the majority (103) occurring in *ethA*. Notably however, most (97/103) MIC-elevating *ethA* variants did not raise the ethionamide MIC above the ECOFF (ETH ECOFF = 2 log2MIC minus baseline). Variants in *fabG1* and *inhA* were common and strongly associated with elevated ethionamide MICs (Fig. S2). Seven resistance-associated variants were identified in the alternative activating enzymes for ethionamide, *Rv3083* (5) and *RvO565c* (2), and three resistance-associated variants were found in the non-canonical ethionamide gene *mshA*. The relative lack of mutations with significant effects identified in the alternate monooxygenases may reflect their decreased relative abundance as a proportion of the total monooxygenase pools of the strains sampled in this study, as found in a previous study, although this was not biochemically verified here[41]. Two mutations in *ethR* were associated with decreased ethionamide MICs, consistent with its role as a regulator of the prodrug-activating enzyme *ethA*.

Resistance to delamanid is mediated by inactivating mutations in *ddn* or by mutations that affect the cofactor $F_{420}$ biosynthesis pathway (namely *fgd1* and *fbiA-D*). We identified eleven mutations in *ddn*, seven in *fbiA*, and one in *fbiC* that were associated with increases in delamanid MIC (Fig. S2, Supplementary Data 3). Over half (6/11) of the mutations in *ddn* were nonsense or frameshift mutations.

### Effect of genetic background on MIC

Several studies have noted that the strain genetic background can influence MICs in addition to primary resistance mutations[36,42,43]. In this study, we found that the effects of lineage on isolate MIC tended to be small compared to primary resistance allele effects for most drugs (mean lineage effect 0.41 log2MIC, mean lineage effect to median primary resistance allele effect ratio 0.15), yet still statistically significant (Fig. S3). Notably however, lineage three was associated with a 1.5 lower moxifloxacin log2MIC compared to lineage four after controlling for primary resistance alleles in *gyrA* and *gyrB*.

### Interactions beyond additivity

We also sought to identify whether there were any effects beyond additivity for co-occurring mutation pairs. Out of 96 pairs tested across 13 drugs, we identified three mutation pairs with greater than additive effects on ethambutol resistance and two pairs with greater than additive rifampicin resistance (Fig. S4, Supplementary Data 10). The interaction of these mutations resulted in log2MICs increased beyond additivity by 1.4–2.4 log2MIC, which resulted in MICs well beyond that of the strongest individual mutations for ethambutol and *rpoB* S450L for rifampicin. Interestingly, we also identified a mutation pair in rifabutin (*rpoB* L430P with *rpoB* D435G) where, in our interaction model, the individual mutations were no longer associated with resistance to rifabutin when occurring individually but are associated with resistance when co-occurring (Supplementary Data 10). These mutations are in sites previously associated with low-level resistance to rifampicin, so it is possible that the combined disturbance to the drug binding site is required to mediate their resistance-causing potential

for rifabutin, although this remains to be experimentally verified. The remaining significant mutation-pairs either consisted of a known resistance mutation with a putative compensatory mutation (such as *rpoB* with *rpoC*) or had additive MICs that were in the tails of the distribution, suggesting that interaction effects were reflecting assay thresholds, at least in part, as opposed to true effects.

### Extension beyond the 2021 WHO catalog

To assess how measurement of MICs improves our ability to detect meaningful genetic associations with resistance/susceptibility, we compared our MIC-based catalog with the recently published 2021 WHO catalog for tuberculosis (Supplementary Data 7)[16]. 179 unique mutation-phenotype associations were found in both catalogs, with nearly a third (59/179) classified as "resistant – interim". Our model finds that 61% (36/59) of these mutations are associated with significant elevations in MIC in our data, of which 14 were sub-ECOFF and therefore unlikely to be confidently identified by binary methods. The inability of binary methods to detect these smaller but significant elevations in MIC is also shown by the lack of associations in *RvO678* for bedaquiline and clofazimine in the WHO catalog, although this is mentioned as a limitation. Notably, we have shown in a separate work that the heritability of resistance for bedaquiline and clofazimine improves dramatically when we detect MICs as opposed to binary phenotypes, consistent with our findings here that many of *RvO678* mutations result in sub-ECOFF elevations in MIC[44].

## Discussion

In this study, we used WGS combined with high throughput MIC measurements to develop a quantitative catalog of resistance to thirteen anti-tuberculosis drugs. Linking mutations to MICs allows for a rapid and reliable alternative to phenotypic DST for individual isolates that does not rely on critical concentrations that may be revised. These results can help to improve diagnostics and guide future study designs trialing high dose therapies of less toxic and more effective drugs (e.g., rifampicin, isoniazid and moxifloxacin)[10,11,25].

Notably, we identified 321 mutations whose effects on MIC are entirely or partially below their respective ECOFF. Further work is needed to understand whether these mutations lead to increased treatment failure and/or relapse rates as is the case for the "borderline" mutations in *rpoB* for rifampicin[28]. If so, rapid molecular assays should be employed to detect these variants.

We also found mutations associated with increased susceptibility to bedaquiline, clofazimine, and aminoglycosides, which raises the intriguing possibility of optimizing regimens based on hypersensitivity as opposed to resistance. Given the relatively common rate of inactivating mutations in *mmpL5*, rapid molecular tests should be developed to ensure that these isolates are not falsely identified as resistant. Deletion of other transcriptional regulators has also been shown to increase bedaquiline susceptibility, suggesting other sensitizing mutations may also occur[45]. Further work to understand the distribution and frequency of these mutations may help elucidate their clinical relevance globally.

Our new catalog was unable to explain most binary resistance to ethionamide, bedaquiline, clofazimine, linezolid and delamanid, implying that many new variants and loci remain to be discovered (Fig. S5)[46]. More widespread use of these drugs clinically will facilitate collection of resistant strains for use in GWAS to identify other genetic loci involved in resistance; however, high levels of inactivating variation were observed in *ethA* (ethionamide), *ddn* (delamanid) and *RvO678* (bedaquiline/clofazimine), suggesting that many isolates will need to be sampled to achieve saturation for these drugs, similar to pyrazinamide. Alternative approaches relying on random mutagenesis, directed evolution, and machine learning have been employed to generate predictions for mutations that have never been observed in a patient, however these may not always identify mutations that are

competitive in vivo[47–54]. The database generated by CRyPTIC can be used as a resource for these approaches by highlighting which mutations actually occur in patients and acting as a training set for machine learning algorithms.

Limitations to this study include the lower number of isolates resistant to newer drugs, the lack of isolates from lineages 5 and 6, which are responsible for a significant proportion of cases in sub-Saharan Africa, potential misattribution of mutational effects outside our target genes or due to exclusion of insertions/deletions >50 bp in size from our model, and the use of ECOFFs that have not yet been extensively validated against other methods, although we have shown good concordance with MGIT and MODS results[17]. In addition, it has been shown that minor alleles at sites associated with resistance can influence MIC[55]. While we have tried to limit this effect by removing isolates for which we could not confidently call a variant at a site previously associated with resistance, it is possible that novel resistance-associated sites with minor alleles could affect our model. We have attempted to limit erroneous associations through controlling for lineage and population structure in our modeling approach as well as by validating mutations through structural mapping and degree of homoplasy where possible. Finally, changes in transcription or translation may also mediate antibiotic tolerance and persistence states to impact the efficacy of antibiotics in vivo[56].

## Methods
### Ethics statement
Approval for CRyPTIC study was obtained by Taiwan Centers for Disease Control IRB No. 106209, University of KwaZulu Natal Biomedical Research Ethics Committee (UKZN BREC) (reference BE022/13) and University of Liverpool Central University Research Ethics Committees (reference 2286), Institutional Research Ethics Committee (IREC) of The Foundation for Medical Research, Mumbai (Ref nos. FMR/IEC/TB/01a/2015 and FMR/IEC/TB/01b/2015), Institutional Review Board of P.D. Hinduja Hospital and Medical Research Centre, Mumbai (Ref no. 915-15-CR [MRC]), scientific committee of the Adolfo Lutz Institute (CTC-IAL 47-J / 2017) and in the Ethics Committee (CAAE: 81452517.1.0000.0059) and Ethics Committee review by Universidad Peruana Cayetano Heredia (Lima, Peru) and LSHTM (London, UK).

### Dataset collection
The CRyPTIC dataset collection and processing has been previously described in detail[17]. Briefly, clinical isolates were sub-cultured before inoculation of a single biological replicate into CRyPTIC-designed 96-well microtiter plates manufactured by ThermoFisher. Plates contained doubling-dilution ranges for 14 different antibiotics (para-aminosalicylic acid was excluded from the study due to poor-quality results on the plate). Isolate MICs were read after 14 days by a laboratory scientist using a Thermo Fisher Sensititre Vizion digital MIC viewing system and an image of the plate was also uploaded to a bespoke web server, allowing for additional MIC measurements by an automated computer vision system (AMYGDA) and by citizen science volunteers (Bash the Bug Zooniverse project) as previously described[57,58]. MIC measurements were classified as high (all three methods agree), medium (only two methods agree), or low (no methods agree). Previous work has shown that using multiple methods catches cases where either the laboratory scientist or software have made an error in calling the MIC[17]. While sequencing processes differed slightly between CRyPTIC laboratories, all sequencing was performed using Illumina. The Clockwork sequence processing pipeline took in paired FASTQ files before filtering, mapping, and providing variant calls for each isolate (Clockwork available from: https://github.com/iqbal-lab-org/clockwork, more detailed description of pipeline available in[16]). Isolates that had both phenotypic and whole-genome sequencing data were used as a starting dataset for this study[17]. ECOFFs were defined in Cryptic Consortium et al 2022 and are provided in Supplementary Data 1[17].

### Target gene selection
Target genes were selected based on the results of a prior study and through a literature search for each drug[59]. Mutations occurring in genes and the 100 bp directly upstream of each gene were considered as candidates for inclusion in this study. Genomic positions for each gene considered (not including 100 bp upstream) are provided from the H37Rv v3 genbank file in Supplementary Data 9.

### Statistical modeling
All genetic variations smaller than 50 bp occurring in the target genes for each drug (Supplementary Data 1) were included as candidates for effects in this study. Large insertions, deletions, and other structural changes larger than 50 bp were not included in this study, due to limitations with the re-genotyping approach employed across all isolates. Both in-frame and frameshifting insertion/deletion (indel) mutations occurred in the dataset; however, only two positions harbored indels of both types (the in-frame deletions being 3 bp and 12 bp in rpoB). As the phenotypes of these isolates carrying in-frame deletions were similar to the frameshifting indels occurring at the same site, these indels were pooled as one candidate effect. Other indel mutations that occurred at the same position (either all in-frame or all frameshifting) were also pooled as one candidate effect to boost statistical power given their likely shared mechanism and size of effect. Mutations that always co-occurred in the dataset were combined into one candidate effect with all mutations named. Isolates were excluded from analysis if they contained evidence for mixed alleles at positions previously associated with resistance to that drug (i.e., a mixed allele call for position S450 in rpoB for rifampicin) to reduce potential instances of hetero-resistant isolates[22]. Interval regression was performed in Stata version 16.1 with a genomic cluster variable as a random effect to control for population structure. Cluster ID was determined by performing agglomerative clustering with complete linkage criterion using Scikit-learn in Python on a whole-genome SNP distance array of all isolates in the dataset[60]. A sensitivity analysis was performed to compare the effects of clustering at 12, 25, 50, and 100 single nucleotide polymorphism (SNP) distances (100 used for all results shown). Lineage and laboratory performing the MICs (SITEID variable) were included as fixed effect, factor variables to control for genetic and technical variation in each individual drug model. MICs were encoded as the interval with upper bound log2(MIC) and lower bound log2(MIC minus 1 doubling dilution). The bottom and top wells were extended by three doubling dilutions to account for censoring. The generalized form of the equation for the interval regression model is below:

$$\text{Censored } \log_2(\text{MIC}) = \mathbf{X_i B} + \mathbf{Z_i u_i} \tag{1}$$

Where $\mathbf{X_i}$ denotes the variable list, with lineage and technical site being fixed, factor variables and all other mutations tested being fixed binary variables. $Z_i$ is the random effects groupings, which were defined by cluster ID. $B$ and $u_i$ denote the calculated fixed and random effects, respectively.

The Benjamini-Hochberg correction was used to adjust raw p-values and the false discovery rate was set at 5% for each drug based on the number of variants considered, including all variants in one mutually adjusted multivariable model. Mutations that have statistically significant effects on log2MIC > 0 are defined as resistance-associated for the purposes of this study. Mutation effect size relative to the ECOFF is noted where relevant. Pairs of mutations that occurred in at least five isolates with each individual mutation occurring at least five times were subsequently tested for interactions in a mixed effect interval regression model containing all other variants for that drug reaching the significance threshold (Benjamini-Hochberg adjusted $p$ value < 0.05).

## Data preparation, analysis, and figure-making

Data was prepared for analysis using Python, statistical outputs were analyzed using R, and figures were made using ggPlot2 in R[61]. Homoplasy was calculated using HomoplasyFinder, with a mutation considered homoplastic if it had evolved in at least two independent occurences[62]. A R file that recapitulates all the post-model analysis and figures is available[61] in the Supplemental material (Supplemental Code). Structural modeling was done using UCSF Chimera[63].

## Reporting summary

Further information on research design is available in the Nature Portfolio Reporting Summary linked to this article.

## Data availability

The ENA IDs, variant call files, and MICs are all available at a permanent ftp site at EMBL-EBI (http://ftp.ebi.ac.uk/pub/databases/cryptic/release_june2022/). This site also includes processed tables with unique IDs that match genotype and phenotype information for facile use.

## Code availability

The Clockwork variant calling pipeline is available from: https://github.com/iqbal-lab-org/clockwork. Scripts used for statistical analysis in Stata and analysis of results in R are available from: https://github.com/carterjosh/cryptic-mic. DOI for Github repository: https://zenodo.org/doi/10.5281/zenodo.10065150[64].

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

## Acknowledgements

J.C. would like to thank Spencer Dunleavy (University of Pennsylvania Medical School, Philadephia, USA). We thank Faisal Masood Khanzada and Alamdar Hussain Rizvi (NTRL, Islamabad, Pakistan), Angela Starks and James Posey (Centers for Disease Control and Prevention, Atlanta, USA), and Juan Carlos Toro and Solomon Ghebremichael (Public Health Agency of Sweden, Solna, Sweden). This work was supported by Wellcome Trust/Newton Fund-MRC Collaborative Award (200205/Z/15/Z); and Bill & Melinda Gates Foundation Trust (OPP1133541). Oxford CRyPTIC consortium members are funded/supported by the National Institute for Health Research Health Protection Research Unit (NIHR HPRU) in Healthcare Associated Infections and Antimicrobial Resistance at Oxford University in partnership with the UK Health Security Agency (NIHR200915), and the NIHR Biomedical Research Centre, Oxford. The views expressed are those of the authors and not necessarily those of the NHS, the NIHR, the Department of Health or the UK Health Security Agency. The funders had no role in study design, data collection and

analysis, decision to publish, or preparation of the manuscript. J.M. is supported by the Wellcome Trust (203919/Z/16/Z). Z.Y. is supported by the National Science and Technology Major Project, China Grant No. 2018ZX10103001. K.M.M. is supported by EMBL's EIPOD3 program funded by the European Union's Horizon 2020 research and innovation program under Marie Skłodowska Curie Actions. T.C.R. is funded in part by funding from Unitaid Grant No. 2019-32-FIND MDR. R.S.O. is supported by FAPESP Grant No. 17/16082-7. L.F. received financial support from FAPESP Grant No. 2012/51756-5. B.Z. is supported by the National Natural Science Foundation of China (81991534) and the Beijing Municipal Science & Technology Commission (Z201100005520041). N.T.T.T. is supported by the Wellcome Trust International Intermediate Fellowship (206724/Z/17/Z). G.T. is funded by the Wellcome Trust. R.W. is supported by the South African Medical Research Council. J.C. is supported by the Rhodes Trust and Stanford Medical Scientist Training Program (T32 GM007365). A.L. is supported by the National Institute for Health Research (NIHR) Health Protection Research Unit in Respiratory Infections at Imperial College London. S.G.L. is supported by the Fonds de Recherche en Santé du Québec. C.N. is funded by Wellcome Trust Grant No. 203583/Z/16/Z. A.V.R. is supported by Research Foundation Flanders (FWO) under Grant No. G0F8316N (FWO Odysseus). G.M. was supported by the Wellcome Trust (098316, 214321/Z/18/Z, and 203135/Z/16/Z), and the South African Research Chairs Initiative of the Department of Science and Technology and National Research Foundation (NRF) of South Africa (Grant No. 64787). The funders had no role in the study design, data collection, data analysis, data interpretation, or writing of this report. The opinions, findings and conclusions expressed in this manuscript reflect those of the authors alone. L.G. was supported by the Wellcome Trust (201470/Z/16/Z), the National Institute of Allergy and Infectious Diseases of the National Institutes of Health under award number 1R01AI146338, the GOSH Charity (VC0921) and the GOSH/ICH Biomedical Research Centre (www.nihr.ac.uk). A.B. is funded by the NDM Prize Studentship from the Oxford Medical Research Council Doctoral Training Partnership and the Nuffield Department of Clinical Medicine. D.J.W. is supported by a Sir Henry Dale Fellowship jointly funded by the Wellcome Trust and the Royal Society (Grant No. 101237/Z/13/B) and by the Robertson Foundation. A.S.W. is an NIHR Senior Investigator. T.M.W. is a Wellcome Trust Clinical Career Development Fellow (214560/Z/18/Z). A.S.L. is supported by the Rhodes Trust. R.J.W. receives funding from the Francis Crick Institute which is supported by Wellcome Trust, (FC0010218), UKRI (FC0010218), and CRUK (FC0010218). T.C. has received grant funding and salary support from US NIH, CDC, USAID and Bill and Melinda Gates Foundation. The computational aspects of this research were supported by the Wellcome Trust Core Award Grant Number 203141/Z/16/Z and the NIHR Oxford BRC. Parts of the work were funded by the German Center of Infection Research (DZIF). The Scottish Mycobacteria Reference Laboratory is funded through National Services Scotland. The Wadsworth Center contributions were supported in part by Cooperative Agreement No. U60OE000103 funded by the Centers for Disease Control and Prevention through the Association of Public Health Laboratories and NIH/NIAID grant AI-117312. Additional support for sequencing and analysis was contributed by the Wadsworth Center Applied Genomic Technologies Core Facility and the Wadsworth Center Bioinformatics Core. SYNLAB Holding Germany GmbH for its direct and indirect support of research activities in the Institute of Microbiology and Laboratory Medicine Gauting. N.R. thanks the Programme National de Lutte contre la Tuberculose de Madagascar. For the purpose of Open Access, the author has applied a CC BY public copyright licence to any Author Accepted Manuscript version arising from this submission.

## Author contributions

Members of the CRyPTIC consortium collected, phenotyped, and sequenced all isolates in the CRyPTIC dataset. J.C., A.S.W., and T.M.W. designed this study, JC performed statistical analyses and structural mapping, J.C. wrote the manuscript, J.C., P.W.F., Z.I., T.E.A.P., T.M.W., and A.S.W. revised the manuscript with all partners providing feedback, and P.W.F., T.M.W.,. A.S.W., and D.W.C. supervised the work.

## Competing interests

E.R. is employed by Public Health England and holds an honorary contract with Imperial College London. I.F.L. is Director of the Scottish Mycobacteria Reference Laboratory. S.N. receives funding from German Center for Infection Research, Excellenz Cluster Precision Medicine in Chronic Inflammation, Leibniz Science Campus Evolutionary Medicine of the LUNG (EvoLUNG)tion EXC 2167. P.S. is a consultant at Genoscreen. T.R. is funded by NIH and DoD and receives salary support from the non-profit organization FIND. T.R. is a co-founder, board member and shareholder of Verus Diagnostics Inc, a company that was founded with the intent of developing diagnostic assays. Verus Diagnostics was not involved in any way with data collection, analysis or publication of the results. T.R. has not received any financial support from Verus Diagnostics. UCSD Conflict of Interest office has reviewed and approved T.R.'s role in Verus Diagnostics Inc. T.R. is a co-inventor of a provisional patent for a TB diagnostic assay (provisional patent #: 63/048.989). T.R. is a co-inventor on a patent associated with the processing of TB sequencing data (European Patent Application No. 14840432.0 and USSN 14/912,918). T.R. has agreed to "donate all present and future interest in and rights to royalties from this patent" to UCSD to ensure that he does not receive any financial benefits from this patent. S.S. is working and holding ESOPs at HaystackAnalytics Pvt. Ltd. (Product: Using whole genome sequencing for drug susceptibility testing for *Mycobacterium tuberculosis*). The remaining authors declare no competing interest.

## Additional information

# The CRyPTIC Consortium

Ivan Barilar[1], Simone Battaglia[2], Emanuele Borroni[2], Angela Pires Brandao[3,4], Alice Brankin[5], Andrea Maurizio Cabibbe[2], Joshua Carter [6] ✉, Darren Chetty[7], Daniela Maria Cirillo[2], Pauline Claxton[8], David A. Clifton[5], Ted Cohen[9], Jorge Coronel[10], Derrick W. Crook[5], Viola Dreyer[1], Sarah G. Earle[5], Vincent Escuyer[11], Lucilaine Ferrazoli[4], Philip W. Fowler[5], George Fu Gao[12], Jennifer Gardy[13], Saheer Gharbia[14], Kelen Teixeira Ghisi[4], Arash Ghodousi[2,15], Ana Luíza Gibertoni Cruz[5], Louis Grandjean[16], Clara Grazian[17], Ramona Groenheit[18], Jennifer L. Guthrie[19,20], Wencong He[12], Harald Hoffmann[21,22], Sarah J. Hoosdally[5], Martin Hunt[5,23], Zamin Iqbal[23], Nazir Ahmed Ismail[24], Lisa Jarrett[25], Lavania Joseph[24], Ruwen Jou[26], Priti Kambli[27], Rukhsar Khot[27], Jeff Knaggs[5,23], Anastasia Koch[28], Donna Kohlerschmidt[11], Samaneh Kouchaki[5,29], Alexander S. Lachapelle[5], Ajit Lalvani[30], Simon Grandjean Lapierre[31], Ian F. Laurenson[8], Brice Letcher[23], Wan-Hsuan Lin[26], Chunfa Liu[12], Dongxin Liu[12], Kerri M. Malone[23], Ayan Mandal[32], Mikael Mansjö[18], Daniela Vicente Lucena Calisto Matias[25], Graeme Meintjes[28], Flávia de Freitas Mendes[4], Matthias Merker[1], Marina Mihalic[33], James Millard[7], Paolo Miotto[2], Nerges Mistry[32], David Moore[10,34], Kimberlee A. Musser[11], Dumisani Ngcamu[24], Hoang Ngoc Nhung[35], Stefan Niemann[1,36], Kayzad Soli Nilgiriwala[32], Camus Nimmo[16], Max O'Donnell[37], Nana Okozi[24], Rosangela Siqueira Oliveira[4], Shaheed Vally Omar[24], Nicholas Paton[38], Timothy E. A. Peto[5], Juliana Maira Watanabe Pinhata[4], Sara Plesnik[22], Zully M. Puyen[39], Marie Sylvianne Rabodoarivelo[40], Niaina Rakotosamimanana[40], Paola M. V. Rancoita[15], Priti Rathod[25], Esther Rhiannon Robinson[25], Gillian Rodger[5], Camilla Rodrigues[27], Timothy C. Rodwell[41,42], Aysha Roohi[5], David Santos-Lazaro[39], Sanchi Shah[32], Grace Smith[14,25], Thomas Andreas Kohl[1], Walter Solano[10], Andrea Spitaleri[2,15], Adrie J. C. Steyn[7], Philip Supply[43], Utkarsha Surve[27], Sabira Tahseen[44], Nguyen Thuy Thuong Thuong[35], Guy Thwaites[5,35], Katharina Todt[22], Alberto Trovato[2], Christian Utpatel[1], Annelies Van Rie[45], Srinivasan Vijay[46], A. Sarah Walker[5], Timothy M. Walker[5,35], Robin Warren[47], Jim Werngren[18], Maria Wijkander[18], Robert J. Wilkinson[30,48,49], Daniel J. Wilson[5], Penelope Wintringer[23], Yu-Xin Xiao[26], Yang Yang[5], Zhao Yanlin[12], Shen-Yuan Yao[24] & Baoli Zhu[50]

[1]Research Center Borstel, Borstel, Germany. [2]IRCCS San Raffaele Scientific Institute, Milan, Italy. [3]Oswaldo Cruz Foundation, Rio de Janeiro, Brazil. [4]Institute Adolfo Lutz, São Paulo, Brazil. [5]University of Oxford, Oxford, UK. [6]Stanford University School of Medicine, Stanford, CA, USA. [7]Africa Health Research Institute, Durban, South Africa. [8]Scottish Mycobacteria Reference Laboratory, Edinburgh, UK. [9]Yale School of Public Health, Yale, MI, USA. [10]Universidad Peruana Cayetano Heredia, Lima, Perú. [11]Wadsworth Center, New York State Department of Health, Albany, NY, USA. [12]Chinese Center for Disease Control and Prevention, Beijing, China. [13]Bill & Melinda Gates Foundation, Seattle, WA, USA. [14]UK Health Security Agency, London, UK. [15]Vita-Salute San Raffaele University, Milan, Italy. [16]University College London, London, UK. [17]University of New South Wales, Sydney, NSW, Australia. [18]Public Health Agency of Sweden, Solna, Sweden. [19]The University of British Columbia, Vancouver, BC, Canada. [20]Public Health Ontario, Toronto, ON, Canada. [21]SYNLAB Gauting, Munich, Germany. [22]Institute of Microbiology and Laboratory Medicine, IMLred, WHO-SRL Gauting, Gauting, Germany. [23]EMBL-EBI, Hinxton, UK. [24]National Institute for Communicable Diseases, Johannesburg, South Africa. [25]Public Health England, Birmingham, UK. [26]Taiwan Centers for Disease Control, Taipei, Taiwan, ROC. [27]Hinduja Hospital, Mumbai, India. [28]University of Cape Town, Cape Town, South Africa. [29]University of Surrey, Guildford, UK. [30]Imperial College, London, UK. [31]Université de Montréal, Montréal, QC, Canada. [32]The Foundation for Medical Research, Mumbai, India. [33]Institute of Microbiology and Laboratory Medicine, IMLred, WHO-SRL, Munich-Gauting, Germany. [34]London School of Hygiene and Tropical Medicine, London, UK. [35]Oxford University Clinical Research Unit, Ho Chi Minh City, Viet Nam. [36]German Center for Infection Research (DZIF), Hamburg-Lübeck-Borstel-Riems, Hamburg, Germany. [37]Colombia University Irving Medical Center, New York, NY, USA. [38]National University of Singapore, Singapore, Singapore. [39]Instituto Nacional de Salud, Lima, Perú. [40]Institut Pasteur de Madagascar, Antananarivo, Madagascar. [41]FIND, Geneva, Switzerland. [42]University of California, San Diego, CA, USA. [43]Univ. Lille, CNRS, Inserm, CHU Lille, Institut Pasteur de Lille, U1019 - UMR 9017 - CIIL - Center for Infection and Immunity of Lille, F-59000 Lille, France. [44]National TB Reference Laboratory, National TB Control Program, Islamabad, Pakistan. [45]University of Antwerp, Antwerp, Belgium. [46]University of Edinburgh, Edinburgh, UK. [47]Stellenbosch University, Cape Town, South Africa. [48]Wellcome Centre for Infectious Diseases Research in Africa, Cape Town, South Africa. [49]Francis Crick Institute, London, UK. [50]Institute of Microbiology, Chinese Academy of Sciences, Beijing, China. ✉e-mail: jjcarter@stanford.edu

