## [Peer Review File · Nature Communications]

Quantitative measurement of antibiotic resistance in
Mycobacterium tuberculosis reveals genetic determinants of
resistance and susceptibility in a target gene approachEditorial Note: This manuscript has been previously reviewed at another journal that is not operating a transparent peer review scheme. This document only contains reviewer comments and rebuttal letters for versions considered at *Nature Communications*.

Reviewer #1 (Remarks to the Author):

The authors have successfully tackled most of my comments, and addressed the main limitations on including more methodological details and on the availability of genomic and phenotypic metadata. I am happy for this work to be published provided the authors address the following remaining queries:

- Regarding my previous comment on detecting hetero-resistance, I don't see the value of not considering all nucleotide positions in DR genes when detecting heterozygous alleles and possible cases of heteroresistance. Limiting this analysis to only "positions previously associated with resistance" risks including minor and novel drug-resistance alleles not detected by the Clockwork pipeline, but potentially associated with MIC changes, reducing statistical power.
- Regarding my previous comment on data availability, I can see this has been included under Methods section "Dataset collection". For clarity, please make sure a 'Data availability' section/statement is included that lists all this information, plus a URL to code repositories.
- In the statement "we concluded that the most likely explanation for the few samples where *ahpC* mutations seemed to have an effect were due to undetected mutations in the canonical resistance genes", the authors should provide evidence of these "undetected mutations", e.g. *katG* with large indels.

Reviewer #2 (Remarks to the Author):

Thanks to the author for the detailed response. Overall the data presented is relevant and informative about mutations that can impact MIC even if in many cases do not lead to clinical resistance as those mutations may have an impact on longer treatment or worse treatment outcomes. In the light of that I just want to add some (minor) additional comments which can help to improve the article:

1. Please unify nomenclature when you refer to ECOFF. Sometimes you refer to ECOFF and other times to CRYPTIC ECOFF (in one of the figures). I assume you always refer to CRYPTIC ECOFF. Please unify terminology to avoid confusion with clinical guidelines from EUCAST and other institutions.
2. Related to ECOFF I am still not sure how you refer to them in the text and how relates to those published in the ERJ paper. For example you refer to AMI ECOFF = 2.3 log₂MIC (line 315) at some point but in ERJ publication AMI ECOFF is 1 ug/mL- please clarify for the reader the use of ECOFF and doublecheck with the ERJ paper given that both publications are somewhat linked
3. When you refer to mutations linked to hypersensitivity in the abstract I think it is safer to say "likely linked". As noticed in one of your responses, mechanistic assays are needed to corroborate the phenotype
4. In several places italics are missed for the gene names and for *Mycobacterium tuberculosis*, *Mtb* - please correct

Reviewer #3 (Remarks to the Author):

Thank you for your considered responses to previous reviewer comments, which clarify important aspects of this work. I believe that this is an important body of work, which will add substantially to our knowledge on genotype-phenotype relationships in MTB, and which can be built upon to develop better tools for individually tailored therapy for DR-TB. I have no further comments on this manuscript.

REVIEWER COMMENTS

Reviewer #1 (Remarks to the Author):

The authors have successfully tackled most of my comments, and addressed the main limitations on including more methodological details and on the availability of genomic and phenotypic metadata. I am happy for this work to be published provided the authors address the following remaining queries:

- Regarding my previous comment on detecting hetero-resistance, I don't see the value of not considering all nucleotide positions in DR genes when detecting heterozygous alleles and possible cases of heteroresistance. Limiting this analysis to only "positions previously associated with resistance" risks including minor and novel drug-resistance alleles not detected by the Clockwork pipeline, but potentially associated with MIC changes, reducing statistical power.

While we sympathize with the reviewer's line of thinking, we believe that throwing out all isolates with a filter fail call anywhere in the gene set is overly conservative, resulting in loss of sample size and hence statistical power. In the Clockwork variant calling pipeline, isolates can have variants fail to be called at a position for several reasons, including minimum fraction read support <90% (potential heterozygous sites), but also minimum and maximum depth of reads. For larger gene sets (either because of large genes such as *rpoB/rpoC* or many included genes), this results in 1000's of codons where a single null or filter fail call would result in removal of the isolate. To demonstrate the effect the more stringent cut-off would have, we calculated a table showing the number of isolates that are removed for each drug either removing only sites with filter fail calls at positions previously associated with resistance or sites with a filter fail call anywhere in the gene.

	Removing all filter fails	Removing fails at resistance-associated sites
DRUG	# Mutations included in model (# Number isolates removed due to filter fail calls)	# Mutations included in model (# Number isolates removed due to filter fail calls)
INH	495 (1369)	544 (142)
EMB	662 (1331)	708 (134)
AMI	430 (1301)	451 (110)
CFZ	361 (716)	377 (0)
RFB	344 (5860)	604 (138)
BDQ	299 (636)	302 (0)

ETH	514 (1537)	563 (48)
MXF	287 (371)	288 (193)
LEV	287 (371)	288 (193)
DLM	263 (1061)	277 (0)
KAN	430 (1301)	451 (110)
RIF	344 (5860)	604 (138)
LZD	105 (846)	109 (8)

The stricter cutoff would result in removal of over 1/3 of isolates for RIF and RFB, hampering our power to investigate the effects of more rare mutations in genes where the sites of resistance mutations have been fairly-well described. While potential effects of heterozygous sites are interesting and indeed appear to have an effect for at least the fluoroquinolones at previously resistance-associated sites [Brankin *et al* 2023 <https://pubmed.ncbi.nlm.nih.gov/37025302/>], investigation of these requires more detailed analysis and deeper sequencing that is beyond the scope of this work. We think that removal of isolates with filter fail calls at sites previously associated with resistance represents the best compromise to minimize effects of heterozygous alleles at likely functional sites without throwing out data that has clean sequencing in relevant regions but a filter fail call in a region of the gene that is highly unlikely to be associated with resistance. We do think that further work to investigate this on a drug-by-drug basis is warranted however and note that this is a limitation of this work in the Discussion section with the following text.

“In addition, it has been shown that minor alleles at sites associated with resistance can influence MIC⁵⁶. While we have tried to limit this effect by removing isolates for which we could not confidently call a variant at a site previously associated with resistance, it is possible that novel resistance-associated sites with minor alleles could affect our model.”

- Regarding my previous comment on data availability, I can see this has been included under Methods section “Dataset collection”. For clarity, please make sure a ‘Data availability’ section/statement is included that lists all this information, plus a URL to code repositories.

Sections for Data and Code availability have been added and include urls to the ftp site with sequencing data and Github repo for code used for statistical modeling and analysis. The added text is below:

“Data Availability:

The ENA IDs, VCFs, and MICs are all available here at a permanent ftp site at EMBL-EBI (http://ftp.ebi.ac.uk/pub/databases/cryptic/release_june2022/).

Code Availability:

The Clockwork variant calling pipeline is available from: <https://github.com/igbal-lab-org/clockwork>. Scripts used for statistical analysis in Stata and analysis of results in R are available from: <https://github.com/carterjosh/cryptic-mic>.”

- In the statement “we concluded that the most likely explanation for the few samples where *ahpC* mutations seemed to have an effect were due to undetected mutations in the canonical resistance genes”, the authors should provide evidence of these “undetected mutations”, e.g. *katG* with large indels.

While the submitted work is based on release one of the CRYPTIC variant database, we accessed a draft version of release two of the CRYPTIC tables, which uses an updated version of the variant calling pipeline Clockwork (v0.12.4 vs v0.8.3). Importantly, in this version, at least 9 isolates for which *ahpC* mutations appeared to have an effect in the model based on release 1, which were now detected to harbor frameshifting mutations, including long indels between 68 and 342 base deletions that would have not been included in the indel calls present in release one. This is a limitation to this work and the below text has been added to clarify the evidence for this statement.

“Mutations in *ahpC* were associated with increased MICs; however, these mutations almost always co-occurred with mutations in canonical isoniazid genes and investigation of the interaction between these co-occurring mutation pairs revealed that *ahpC* mutations did not result in additive resistance, consistent with their proposed compensatory role (Figure 4A). Further investigation of these apparent discrepant isolates using an improved version of the Clockwork variant calling pipeline that detected deletions larger than 50bp identified 9 isolates with apparent resistance-associated *ahpC* mutations that actually harbored large deletions in *katG* not reported in the original variant set used for the model. Thus, the apparent effect of these mutations is likely due to isolates with undetected mutations in the canonical resistance genes as opposed to a bona fide individual effect on isoniazid MIC by mutations in *ahpC*.”

Reviewer #2 (Remarks to the Author):

Thanks to the author for the detailed response. Overall the data presented is relevant and informative about mutations that can impact MIC even if in many cases do not lead to clinical resistance as those mutations may have an impact on longer treatment or worse treatment outcomes. In the light of that I just want to add some (minor) additional comments which can help to improve the article:

1. Please unify nomenclature when you refer to ECOFF. Sometimes you refer to ECOFF and other times to CRYPTIC ECOFF (in one of the figures). I assume you always refer to CRYPTIC ECOFF. Please unify terminology to avoid confusion with clinical guidelines from EUCAST and other institutions.

Thanks for catching this discrepancy. The figures that referenced CRYPTIC ECOFFs/CCs have been updated to ECOFF.

2. Related to ECOFF I am still not sure how you refer to them in the text and how relates to those published in the ERJ paper. For example you refer to AMI ECOFF = $2.3 \log_2 \text{MIC}$ (line 315) at some point but in ERJ publication AMI ECOFF is 1 ug/mL- please clarify for the reader the use of ECOFF and doublecheck with the ERJ paper given that both publications are somewhat linked

Model effects are relative to a “baseline” MIC (i.e. the constant Y-intercept of the interval regression model). To facilitate comparison with the ECOFFs, we subtract this baseline MIC value from the ECOFF MIC to understand whether a individual mutation would be capable of increasing the MIC from baseline to above the ECOFF. Thus, where we compare to ECOFFs in the text, we are actually comparing the effect an isolate would have to have to raise the MIC above the ECOFF from baseline. This is an important clarification, so thank you for raising the confusion. To clarify this, we have added the following text to the beginning of the results section. We also note that all ECOFFs are minus the baseline MIC where relevant.

“To facilitate comparison with the previously published ECOFF values, we report mutational effects relative to the difference between the ECOFF MIC and the baseline MIC calculated by the model for each drug. Thus, if a mutation is associated with an effect larger than the ECOFF minus baseline, it is associated with an increase in resistance that would be above what is considered wildtype on the plate.”

3. When you refer to mutations linked to hypersensitivity in the abstract I think it is safer to say “likely linked”. As noticed in one of your responses, mechanistic assays are needed to corroborate the phenotype

We have changed the relevant text in the abstract to the following:
“This identified 449 unique MIC-elevating variants across thirteen drugs, as well as 91 mutations likely linked to hypersensitivity.”

4. In several places italics are missed for the gene names and for Mycobacterium tuberculosis, Mtb - please correct

Proper italicization has been checked and added where appropriate.

Reviewer #3 (Remarks to the Author):

Thank you for your considered responses to previous reviewer comments, which clarify important aspects of this work. I believe that this is an important body of work, which will add substantially to our knowledge on genotype-phenotype relationships in MTB, and which can be built upon to develop better tools for individually tailored therapy for DR-TB. I have no further comments on this manuscript.

Reviewer #1 (Remarks to the Author):

The authors have successfully addressed my few remaining comments including a good justification of removal of isolate genomes based on heterozygous alleles at sites previously associated with resistance; new section on Data and Code availability, and clarification on the role of *ahpC* mutations.